# Sensing Optimum in the Raw: Leveraging the Raw-Data Imaging Capabilities of Raspberry Pi for Diagnostics Applications

**DOI:** 10.3390/s21103552

**Published:** 2021-05-20

**Authors:** Alessandro Tonelli, Veronica Mangia, Alessandro Candiani, Francesco Pasquali, Tiziana Jessica Mangiaracina, Alessandro Grazioli, Michele Sozzi, Davide Gorni, Simona Bussolati, Annamaria Cucinotta, Giuseppina Basini, Stefano Selleri

**Affiliations:** 1DNAPhone S.R.L., Viale Mentana 150, 43121 Parma, Italy; alessandro.tonelli@dnaphone.it (A.T.); Veronicamangia93@gmail.com (V.M.); alessandro.candiani@dnaphone.it (A.C.); francesco.pasquali@dnaphone.it (F.P.); jessica.mangiaracina@dnaphone.it (T.J.M.); alessandro.grazioli@dnaphone.it (A.G.); michele.sozzi@dnaphone.it (M.S.); 2H&D S.R.L., Strada Langhirano 264/1a, 43124 Parma, Italy; d.gorni@hedsrl.it; 3Dipartimento di Scienze Medico-Veterinarie, Via del Taglio 10, 43126 Parma, Italy; simona.bussolati@unipr.it (S.B.); giuseppina.basini@unipr.it (G.B.); 4Dipartimento di Ingegneria e Architettura, University of Parma, Parco Area delle Scienze, 181/A, 43124 Parma, Italy; annamaria.cucinotta@unipr.it

**Keywords:** Raspberry Pi, CMOS, raw data, photometric analysis, imaging, oxidative stress, d-ROM Fast, PAT, horse blood

## Abstract

Single-board computers (SBCs) and microcontroller boards (MCBs) are extensively used nowadays as prototyping platforms to accomplish innovative tasks. Very recently, implementations of these devices for diagnostics applications are rapidly gaining ground for research and educational purposes. Among the available solutions, Raspberry Pi represents one of the most used SBCs. In the present work, two setups based on Raspberry Pi and its CMOS-based camera (a 3D-printed device and an adaptation of a commercial product named We-Lab) were investigated as diagnostic instruments. Different camera elaboration processes were investigated, showing how direct access to the 10-bit raw data acquired from the sensor before downstream imaging processes could be beneficial for photometric applications. The developed solution was successfully applied to the evaluation of the oxidative stress using two commercial kits (d-ROM Fast; PAT). We suggest the analysis of raw data applied to SBC and MCB platforms in order to improve results.

## 1. Introduction

In the past decade, mobile phones, microcontroller boards (MCBs) and single-board computers (SBCs) gained increasing attention as imaging, sensing and diagnostics tools [1,2,3,4,5].

The fast-growing availability of microcontrollers (such as Arduino, Trinket or Propeller mini) and single-board computers (such as Raspberry Pi) is inspiring applications completely outside topics related to computer sciences. The easy access to open-source software, hardware and to high performance optoelectronic and robotic components for these universal electronic modules is pushing the application in laboratory-related fields for research, customization, prototyping and educational purposes.

Some peculiar aspects of these platforms fit with the needs in biological and chemical analysis and more in general in laboratory applications: (i) numerous and flexible input/output signal routing; (ii) numerous options for connectivity with modules for motion, sound, sensing, light, etc.; (iii) on-board availability of numerous standard communication protocols (e.g., USB, ethernet, Wi-Fi, Bluetooth, TX/RX, I^2^C, SPI) that enable easy integration with other devices; (iv) maturity of high-level programming languages that facilitate the programming, the project porting and reduce the learning curve for non-programmers and beginners; and (v) a large community of enthusiasts and open-source oriented programmers that allow for rapid problem solving and self-learning.

According to this scenario, the single-board computer Raspberry Pi plays a relevant role. This low-cost, credit-card-sized computer was developed by the Raspberry Pi foundation with the main aim to foster computer science and STEAM education for children and adults [6]. Currently, the Raspberry Pi catalogue offers several different boards that share the same basic technical concepts, numerous accessories and add-on boards that enable one to accomplish specific tasks and extend the Raspberry Pi capabilities. The offer of several “official” camera modules combined with the available mature software libraries make Raspberry Pi particularly suitable for light detection and artificial vision applications. Moreover, the Linux-based Operative System (OS) and the integration with several programming languages (e.g., Python, JavaScript, C, Ruby) enable massive capabilities for the creation of custom functions and interfaces.

Recently, Raspberry Pi boards have been employed for a wide spectrum of chemical, biological and clinical research applications. Relevant implementations include: the creation/prototyping of devices such as real-time polymerase chain reaction (PCR) [7,8] and digital PCR [9] readers, microfluidic biosensors for pathogens [10], surface plasmon resonance (SPR) spectrometers [11], multi-well plate readers [12], handheld UV fluorescence spectrophotometer readers [13], lab-on-a-chip [14], atomic force microscopes [15], microscopy [16,17], laboratory automation [18,19], microfluidics [20,21], and plant phenotyping [22]. Besides diagnostics, Raspberry Pi-based imaging could also be beneficial in several distant fields [23,24,25,26].

Electrochemical and optical bio/chemosensoristic devices need continuous improvement in terms of integration, portability, cheapness, simplification of experimental protocols, and development of high-throughput approaches [27]. Moreover, characteristics such as connectivity and cloud data management, the customization of consumer electronic components and the integration of open-source 3D printing are rapidly gaining significance [28,29]. Some of those features can play a pivotal role in the development of applications in specific contexts such as resource-poor settings [30], exotic pet medicine [31] or for some niche applications of point-of-care (POC) diagnostics [32].

An emerging topic in veterinary medicine is the control of inflammatory and oxidative stress [33]. These processes seem to be involved and show correlation with numerous pathological conditions, and their monitoring can support evaluations in many fields such as dietary supplementation [34,35], the treatment of chronic conditions [36], heart failure [37], neoplasia [38] and mastitis treatment [39]. Interestingly, despite the fact that the common settings for this kind of analysis often involve *on site* diagnostics, there is still a lack of low-cost portable devices and easy to handle protocols for this purpose.

One of the major benefits for the Raspberry Pi camera module consists of its capability to control all the parameters involved in image acquisition; moreover, the programming of these features through a pure Python interface [40] allows the implementation of complex procedures in a very easy way.

In the present work, different approaches for data acquisition using the Raspberry Pi camera have been investigated and compared. We show that the usage of 10-bit raw data can be beneficial for diagnostic applications, and we tested the developed system using commercial kits for oxidative stress determination. In this paper, a comparison of different methods for diagnostic purposes using only one SBC and a camera sensor as a signal acquisition/elaboration system is presented. The implementation of raw data analysis is proposed as an improving approach and the feasibility in real settings with real samples (oxidative stress evaluation in horse blood) was tested.

According to the presented data, we believe that the implementation of future diagnostic applications on SBCs using raw data imaging could boost diagnostic applications.

## 2. Materials and Methods

### 2.1. Raspberry Pi Basic Settings and Programming

The SBC Raspberry Pi 3 Model B was employed as a computing unit and its first version of the camera module (PiCam, v1.3 board) was employed as a sensor for all the experiments.

The device was installed with the standard Linux-based Raspbian 8 (Jessie) operating system (kernel 4.1.20-v7+) and the experiments were carried out connecting common laptops to the Raspberry Pi-based device through the standard SSH protocol.

The software employed during the experiments was developed using Python (ver. 2.7.9) and the camera module was controlled using the Picamera package (using Version 1.9 as reference documentation), a pure python interface released under BSD license. The developed Python scripts were run directly from the Linux Terminal and the results were saved by the software as csv data and elaborated separately using Excel or R.

The software module that manages image acquisition was created in order to sequentially perform three different data analyses: (i) Auto_mode (AM): exposure and white balance were set as automatic function. In this configuration, the parameters for the image acquisition were managed by the PiCam according to the manufacturer’s settings; (ii) Manual_mode (MM): automatic selection of the exposure and auto-white balance and the application of any kind of effects were disabled; (iii) raw_mode (RM): raw Bayer data were acquired, excluding the de-mosaicing and the post-processing algorithms. For all three approaches, a warm-up time (5 s) for the camera circuitry was declared.

The AM and MM resulted in 8-bit images that automatically underwent cropping and RGB value (0–255) extrapolation by the script, using conventional Python libraries (mainly: Image; numpy; matplotlib). Since the image processing was bypassed using the RM, the output consisted of a numeric matrix reporting the 10-bit RGB values (0–1023) of the corresponding pixels.

Basically, the core of the software function that controls the camera has the following structure:
    def image_processing(camera_framerate, camera_led, camera_image_effect, camera_exposure_mode, camera_exposure_compensation, camera_awb_mode, camera_awb_gains_1, camera_awb_gains_2, camera_shutter_speed, camera_h_flip, camera_v_flip):     stream = io.BytesIO()    with picamera.PiCamera() as camera:        time.sleep(5)        camera.framerate = camera_framerate        camera.led = camera_led        camera.image_effect = camera_image_effect        camera.exposure_mode = camera_exposure_mode        camera.exposure_compensation = int(camera_exposure_compensation)        camera.awb_mode = camera_awb_mode        camera.awb_gains = (float(camera_awb_gains_1), float(camera_awb_gains_2))        camera.shutter_speed = int(camera_shutter_speed)

The class picamera.PiCamera is called along with its attributes relevant for the analysis. The specific values were set outside the function and called depending on the selected imaging method. The class supports the context manager protocol to make this particularly easy (upon exiting a “with” statement, the close() method is automatically called).

The most important parameters, along with their settings, are reported in Figure 1.

The image processing algorithm was explained in our previous paper [41], and the logical scheme is basically the same. Briefly, the reading of samples was controlled by an acquisition module that acquired the image at full resolution (2592 × 1944) from the camera with optimized parameters, cropped it and calculated the mean values for RGB pixel data. The employed ROI was selected through a script based on the Python Image Library (PIL) that acquired a picture of the LED source, identified a small (20 × 20) region with the highest intensity and defined a pixel distance from that center (length and width) in order to have local intensity values ≥ of 95% of the center. The ROI definition was performed once, immediately after the device assembly, and the calculated coordinates were kept constants for all the subsequent experiments.

### 2.2. Device Assembling and Customization

Two similar setups were used for the measurements. Some preliminary characterizations were carried out using a simple 3D-printed setup; basically, this configuration consisted of a holder that allows the correct positioning/alignment of the PiCam, an LED and a plastic light diffuser. The main advantage for this configuration was the possibility to rapidly change LED and diffuser types.

The other employed configuration was based on a commercial product dedicated to STEAM education, named We-Lab (DNAPhone srl, Parma). This configuration is based on an industrially molded plastic case that fit the Raspberry Pi and the PiCam. The system has magnetic modules that, once connected, turn the main module into a microscope or into a colorimeter using the appropriate camera imaging processes controlled by a dedicated Android APP. This setup was employed without substantial hardware modifications for the oxidative stress analyses. However, the standard SD card provided with the commercial version of the device was changed in order to run the custom Python scripts.

The two setups were very similar in terms of component distances, disposition and geometry. Basically, the 3D-printed setup mimicking the internal shapes of We-Lab was employed for characterization purposes and the commercial product was used for real analyses, where a complete device was expected.

All the 3D prototypes were designed and fabricated in-house by means of a 3D printer Sharebot Next Generation (Sharebot, Nibionno, LC, Italy) by polylactic acid (PLA) polymers, a plastic material made from vegetable fibers.

### 2.3. Sample Preparation

The samples for the preliminary characterizations were obtained by dilutions of the food dye “amaranth” (IUPAC name: Trisodium (4E)-3-oxo-4-[(4-sulfonato-1-naphthyl)hydrazono]naphthalene-2,7-disulfonate) in double distilled water. The absorbance was measured by using a Cary 60 spectrophotometer (Agilent Technologies, Santa Clara, CA, USA). The stock solution of the dye was dissolved starting from powder, and all the samples were prepared in order to have absorbance comprised between 0.1 and 1.0 Absorbance units.

The real samples were treated as follows: each horse blood sample was taken from the left jugular vein using a syringe with an 18 G needle. Afterwards, the blood was collected into test tubes containing a coagulation activator for the serum. The samples were stored at 4 °C and were delivered to the laboratory within 12 h of collection. After a centrifugation at 5500 rpm for 4 min, serum and plasma were then divided into aliquots and stored in a deep refrigerator at −18 °C.

### 2.4. Reactive Oxygen Metabolites (ROMs) Test

The ready to use d-ROM Fast Test (H&D srl, Parma, Italy) was employed according to the manufacturer’s instructions. Briefly, 10 μL of R3 solution was added to the pre-filled tube (R2) and mixed by inversion for 10 s. Then, 50 μL of sample was added, mixed by inversion for 15 s and transferred to the pre-filled cuvette (R1) provided by the kit. The cuvette was placed into a FRAS5, a dedicated 37 °C controlled photometer (H&D srl, Parma), and the analysis was carried out selecting the available d-ROM Fast test method from the touch screen of the instrument. The standard method available in the device read a first absorbance (A1) after one minute and a second absorbance value after 1.5 min (A2). The version of FRAS5 used provided the final results along with the corresponding absorbance (A2-A1). For kinetic analysis, each point was acquired manually.

The same samples (same cuvettes) were rapidly transferred to the Raspberry Pi device and analyzed.

### 2.5. Plasma Antioxidant Test (PAT)

The ready to use PAT test (H&D srl, Parma) was employed according to the manufacturer’s instructions using the FRAS5 selecting the test method from the touch screen of the instrument. Briefly, 40 μL of R2 solution was added to the pre-filled cuvette and mixed by inversion for 10 s. After the first absorbance value (A1) was acquired, 20 μL of sample was added and the cuvette was incubated at 37 °C in the instrument for 60 s before reading the second absorbance value (A2). For kinetic analysis, each point was acquired manually.

The same samples (same cuvettes) were rapidly transferred to the Raspberry Pi device and analyzed.

### 2.6. Absorbance Calculation

The absorbance calculation using Raspberry Pi was obtained by image processing for the AM and the MM and by direct manipulation of a numeric matrix for RM. All the images were acquired at a maximum resolution of 5 megapixels (2592 × 1944) and then saved on the microSD card during the assay for further processing. For analysis with the AM and the MM approaches, the same region of interest (ROI) of 30 × 30 pixels was cropped from the original images and the RGB values for each pixel were stored in a matrix by the python script.

The stored data were used for the calculation of mean RGB values. The absorbance (A) was calculated from the mean green (G) channel values considered as intensity, according to:$$A={\;\log}_{10}\frac{\left(I_{0}-I_{d}\right)}{(I_{s}-I_{d)}}$$
where *I**o* is the input light intensity of the LED source, *I_d_* is the residual signal after the LED switching off and *I_s_* is the intensity measured after the sample introduction. Results were saved as a plain text file. The residual signal (*I_d_*) came from the camera once all the light sources were removed, and it was considered as total signal noise for the absorbance calculation.

For the RM approach, the whole numeric matrices for each of the four channels (red, green1, green2, blue) were saved as plain text and simultaneously a matrix sub-set with the same centroid of the cropped images was extrapolated for each sample. Mean values from green channels were employed for absorbance calculations.

The absorbance values obtained with the oxidative stress kits were corrected for the difference in wavelength reading. The peak of the source of We-Lab was centered around 522 mn (spectrum acquired using an USB 4000, Ocean Optics, Dunedin, FL, USA), whereas the FRAS5 instrument performed the reading around 505 nm.

## 3. Results and Discussion

### 3.1. Device Implementation

Raspberry Pi-based devices have a strong potential for analytical applications. We previously investigated this approach using a custom made 3D-printed device in order to accomplish an end-point analysis, the diphenylpicryl-hydrazyl (DPPH) test on samples of bottled tea [41]. In this work, some manual settings for the image acquisition were exploited. However, the raw data were not employed because the required calculations seemed to be too demanding for the used Raspberry Pi model (ver1 B+) combined with the selected type of assay. Moreover, the extracted matrices were quite huge (more than 30 Mb for each channel), making the elaboration and the storage difficult at that time.

One of the most interesting aspects of Raspberry Pi is the capability offered by the software control of the camera module. The employed PiCam is a back-illuminated complementary metal-oxide semiconductor (CMOS) sensor based on the Omnivision OV5647 component. The OV5647 is a 1.4″, 1.4 µm pixel size image sensor with a resolution of 5 megapixels (active array size: 2592 × 1944) with support for 8- and 10-bit raw RGB data output formats. This sensor was conceived primarily for mobile phone and PC multimedia markets and is mounted on the PiCam using an industry standard 8.5 × 8.5 × 5 mm^3^ camera module. The most important parameters of the camera module, such as the white balance, the exposure control and the shutter speed time, can be controlled through a Python library (picamera) as well as the acquisition of the raw Bayer data.

According to the official Picamera documentation, the raw Bayer data differ considerably from simple unencoded captures; they are the data recorded by the camera’s sensor prior to any GPU processing (auto white balance, vignette compensation, smoothing, down-scaling, etc.). The Bayer data structure of the 10-bit values is organized as four 8-bit values, followed by the low-order 2-bits of the four values packed into a fifth byte. The Bayer data in the PiCam are organized in a BGGR pattern (a minor variation of the common Bayer CFA) [40].

In order to investigate the benefits of these control capabilities for diagnostics applications, three data acquisition/elaboration functions were created and executed as needed during sample analysis: (i) Auto_mode (AM). Basically, the default image acquisition mode of the camera module. In this configuration, the automatic exposure control (AEC) and the automatic white balance (AWB) were switched on and the manufacturer image processing algorithms were applied without any control. The output of this process consisted of a “conventional” 8-bit RGB image that was further elaborated with common imaging approaches. (ii) Manual_mode (MM). The AWB and the AEC were disabled, and the exposure compensation and the blue and red channel gains were set as fixed values. Moreover, the image effect option was set to “none”. This approach resulted in a “greenish tones” 8-bit RGB image treated in the same way as AM. This approach is suggested by the camera manual in order to acquire consistent images in terms of brightness, color, and contrast (such as in timelapse photography). (iii) Raw_mode (RM). In this approach, the camera settings were the same as the MM. However, the 10-bit pixel intensities from the OV5647 sensor were directly captured to a file-like object (an io.BytesIO stream) and treated as a numeric matrix. Values from each color channel were grouped without losing the spatial coordinates of the sensor pixel array. At the end of this process, four matrices for each sample were obtained: one from the red channel, one from the blue channel and two from the green channels (Figure 2). The acquired values represent the corresponding pixel signals that come from the sensor before the application of any manufacturer image construction algorithms.

In order to characterize the different approaches, two simple setups were employed (Figure 3).

The setup used for the preliminary characterization was a 3D-printed holder without mobile parts designed in order to fit the PiCam, a light diffuser and an LED, whereas the setup employed for the kit analyses was based on a commercial product, “We-Lab” (DNAPhone srl, Parma), conceived for STEAM education purposes in school settings. The only hardware customization made on the We-Lab consisted of a modified cuvette holder designed to read a lower area of the cuvette provided with the employed diagnostic kits: the final reaction volume was too low to be read for the regular We-Lab.

Both the setups had the PiCam directly mounted into the specific board socket and the LED connected to the same appropriate General Purpose Input/Output (GPIO) pins controlled with the Python-wrapped “wiringpi2” or the “rpi_ws281x” libraries. Since Raspberry Pi does not provide a Digital to Analog Converter (DAC) interface for generating analog voltage for setting the LED intensity, the proper light intensity of the source was achieved via pulse-width modulation (PWM) digital control, acting on the 10-bit (0–1023) duty cycle. LED was directly wired in the 3D-printed setup, whereas the We-Lab provided a magnetic connector that allows the easy plug of the photometer module to the main unit. The 3D-printed setup was designed in order to mimic the internal geometry and volumes of the We-Lab.

Both the setups were controlled using regular laptops (running Linux, Windows or MacOS operative systems depending on the model) connected to the devices using a standard *ssh* connection: the 3D-printed setup was wired through the SSH port, whereas the We-Lab provided an access-point (AP) mode, allowing for a wireless connection on a device’s local network through the standard Wi-Fi network protocol. All the tests were performed remotely, running Python scripts from the connected laptop.

A schematic diagram of the whole analysis process is shown in Figure 1.

The common digital imaging processes usually involve several steps: when photons strike the sensor pixels, signals undergo an analog-to-digital conversion (ADC) step. Then, several processing events occur: the most relevant transformations for this work are the gain control, the de-mosaicing, the color mapping, the white balancing, the gamma correction, the denoising, the sharpening and the compression. In fact, the final stored image is the result of all these calculations.

Usually, the image processing phase occurs through a GPU or dedicated chips that apply specific, and often proprietary, algorithms.

To the best of our knowledge, the multimedia coprocessor on the Broadcom System on Chip (SoC) and the image signal processor (ISP) of the Raspberry Pi rely on proprietary and closed sources and cannot be directly accessed by users. For this reason, the Raspberry Pi foundation has recently released a new open-source camera stack that allows for better access to the internals of the camera system called libcamera [42].

The present work was carried out with the standard camera sensor control and explored the access to the 10-bit raw data as an effective way to minimize the undesired effect coming from the closed image processing steps. In particular, we observed that the automatic camera exposure mode and that the camera auto white balance mode were the major source of variability if set to auto option. Several experiments set in auto mode led to some outlier values along the tested range (data not shown). A possible reason is that depending on the intensity of the LED that comes out from the sample, along with its slightly different profile (size and shape of the spot), the camera data undergo some non-linear calculations in the processing phase steps. Interestingly, some correction of non-linear response (due to gamma correction) using raw data has been previously observed with similar systems applied to other fields [43].

### 3.2. Setup Characterization

Despite the vast PiCam control capabilities, only few works seem to employ its raw image data for research purposes [16,43], and none of them are focused on photometric applications for diagnostics.

In order to evaluate the signal response of the device in the presence of different concentrations of a given molecule, several dilutions of the food dye amaranth were analyzed. The molecule was selected for the similar absorption maximum of the oxidative stress kits tested (520 ÷ 530 nm) and prepared using double distilled water (ddH_2_O) in order to have absorbance values between 0.1 and 1.0 (the starting solution was around 0.87). For each sample, three independent readings for the AM, MM, and RM approaches were performed (Figure 4).

According to the results, the AM and MM performed similarly in terms of absorbance signals, whereas using the RM system the absorbance was closer to values of a conventional spectrophotometer (around 90% of the signal). Moreover, the analysis of the replicas showed that the RM was the more precise approach. The different relative coefficient of variation profiles indicate different behaviors in the signal interpretation for the three approaches: constant for the RM, less precise on low concentration values for the MM and more variable for the AM. Interestingly, the constant tract in the middle concentrations could reflect the effect of some imaging algorithms applied by the camera set in auto mode.

These data suggested that using the RM could be a preferential choice, especially for certain real settings (i.e., on-site analysis or low sample amounts) where obtaining replicates is not often possible.

### 3.3. Reactive Oxygen Metabolites (ROMs) Assay Characterization

The evaluation of the reactive oxygen metabolites in blood is an interesting analytical task for humans and animals. In order to evaluate the capability of the device based on We-Lab to perform this kind of analysis, a commercial kit, the d-ROM Fast Test (H&D srl, Parma), was employed according to the manufacturer’s instructions. A standard sample provided with the kit was used for the analysis and the cuvette was subsequently read with the dedicated instrument FRAS5 (H&D srl, Parma) and with the We-Lab.

The default method of the FRAS5 consists of a kinetic analysis with a total time of less than 2 min. This timeframe was not sufficient for developing a signal to be correctly detected by the We-Lab (data not shown) because of the low signals involved (between 0.01 and 0.08 Abs), combined with the reduced sensitivity of the device compared with the FRAS5.

In order to characterize a suitable tract of the reaction to be read with the We-Lab, a kinetic profile was acquired (Figure 5). The results showed that the RM approach was the most similar in terms of absolute absorbance values and in terms of profile. Moreover, both the FRAS5 and RM plots appeared to be linear and with almost the same slope in the region between 0 and 10 min: this also allows for a kinetic approach with a shifted timeframe (e.g., A1 at five minutes, A2 at 10 min) for the We-Lab using the RM. On the other hand, the not perfectly linear behavior of the AM and the MM approaches suggests more difficulties for the kinetic analysis in the presence of very low absorbance increments.

Additionally, the correlation with the absorbance values from the FRAS5 seemed to be better using the RM (Figure 5b), suggesting that the AM and MM could be not linear.

Interestingly, We-Lab was able to measure high values of absorbance (around 1.8), even if the divergence with FRAS5 was slightly higher in the upper region. Values after 40 min were close to the reaction plateau and were considered acceptable for an end-point version of the assay.

### 3.4. Plasma Antioxidant Assay Characterization

As a further evaluation of the Raspberry-based device capabilities, a second kit for the oxidative stress analysis was tested. For this purpose, the PAT test (H&D srl, Parma) was used according to the manufacturer’s instructions. Adopting a similar approach to dROM Fast Test, a standard sample provided with the kit was used for the analysis. The same cuvette for each sample was read with the dedicated instrument FRAS5 (H&D srl, Parma) and with the We-Lab (Figure 6).

An interesting aspect for this analysis consists of its decreasing profile. This kit was employed in order to evaluate the We-Lab capability to manage this type of reaction. The results showed that, in this case, the RM was also the most suitable approach in terms of absorbance signals. The difference with the FRAS5 is in the order of 10% of the absolute photometric values and is coherent with the data observed in Section 3.2 on the amaranth dye.

A relevant aspect for the We-Lab device derived by the kinetic experiments was its capability to consistently read samples for quite long times (more than 60 min). This evidence supports the employment of a device based on Raspberry Pi and the PiCam for kinetic analysis or for assays that require long incubation times. This feature could be particularly relevant for *in field* applications in the veterinary sector.

### 3.5. Proof of Principle for a POC Application Using Real Sample

In order to evaluate the applicability of a Raspberry Pi-based device such as We-Lab for real-setting diagnostic applications, a session of analysis employing the d-ROM Fast kit on horse blood samples was carried out. In order to allow the reading of all samples on different devices, the default FRAS5 program was chosen, and the We-Lab was programmed and calibrated in order to perform the same analysis as an end-point reaction (Section 3.3). Since the FRAS5 analysis is very fast (in the order of two minutes) and the reaction signal increases rapidly in that region, the end-point approach was regarded as the most robust option to adopt.

The We-Lab device was calibrated with a three-point calibration curve in order to cover the dataset range (values around 30, 130 and 300). All the other concentrations were derived from the calculated linear calibration curve and expressed as Carratelli Units (CARR.U.).

The results on 16 horse blood samples showed good concordance with the method (FRAS5) used as gold standard (Figure 7). According to the previous results, for this experiment only the MM approach was selected from the script in order to minimize the required computational and data storage operations during the session.

The We-Lab device was controlled using a conventional laptop (MacOS operative system) through a Wi-Fi connection. The capability of wireless control as well as the possibility to power the unit with a USB portable power supply or a USB port of a computer represent noteworthy features for POC applications.

The observed data suggest the feasibility of the implementation of Raspberry Pi-based devices equipped with a CMOS camera for diagnostics applications in the POC context. On the other hand, We-Lab, compared with an optimized instrument such as FRAS5, lacks the required optical performance needed for reading the reaction in very short times. Moreover, as a commercial product, FRAS5 offers an integrated solution including a micro-centrifuge in the chassis of the instrument and all the required consumables. Despite not being comparable to this solution, devices such as We-Lab could represent a valid “development” solution in research and education contexts [44], or could open new possibilities for applications in resource-poor countries.

## 4. Conclusions

In this work, a Raspberry Pi-based device coupled with its CMOS camera was successfully used as a reader for photometric analysis. To the extent of our knowledge, this is the first time that a comparison among different imaging approaches was carried out for diagnostic purposes in this field.

According to the obtained results, performing direct analysis of raw 10-bit values directly from the CMOS sensor before any downstream imaging processes could be beneficial for this kind of application. In particular, this solution could improve the performance in prototyping diagnostic devices based on consumer electronics such as SBCs and MCBs. In this work, both a 3D-printed setup and an adaptation of a commercial product based on Raspberry Pi were shown to be good solutions.

In particular, the tested approach was able to read quite high absorbance signals (up to Abs around 1.8), was sufficiently stable to assure a consistent reading on a 60-min timeframe and was successfully operated through a wireless network and without the need for a mains power supply.

This solution was successfully employed with two commercial kits for the evaluation of the oxidative stress (d-ROM Fast and PAT) and for the analysis of the reactive oxygen metabolites in horse blood samples.

Although not conceived for professional usage, this approach could open exciting possibilities for research and education purposes, lowering the costs for accessing affordable instruments and encouraging and strengthening the interactions among different science areas (informatics, biology, chemistry, 3D printing, medicine, etc.). Moreover, in combination with a mobile APP and/or a cloud service, this solution could allow some improvement in limited resource settings.

## Figures and Tables

**Figure 1 sensors-21-03552-f001:**
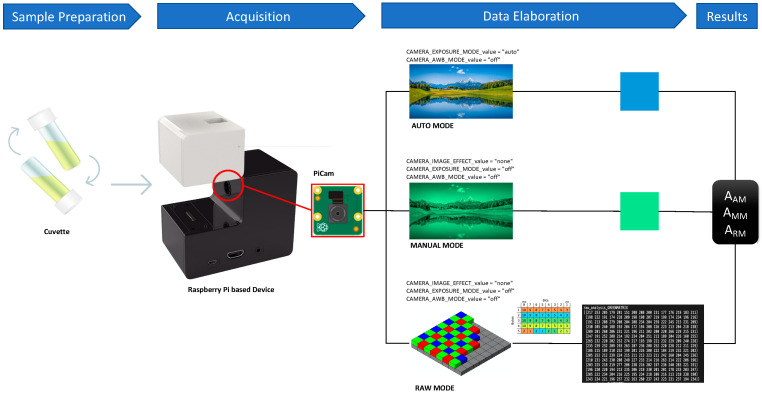
The schematic diagram of the analysis process of the Raspberry Pi-based devices. Briefly, after sample preparation and chemical reactions (where needed), the cuvette was placed in the photometric module and a script launched from a Wi-Fi-connected computer executed the selected approach.

**Figure 2 sensors-21-03552-f002:**
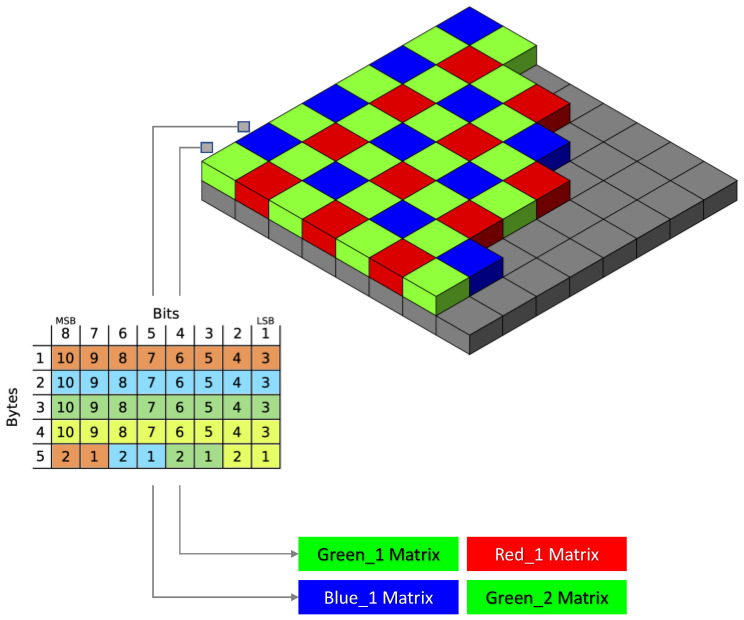
Schematic diagram of Bayer filter of the PiCam CMOS and bits disposition used for data extrapolation.

**Figure 3 sensors-21-03552-f003:**
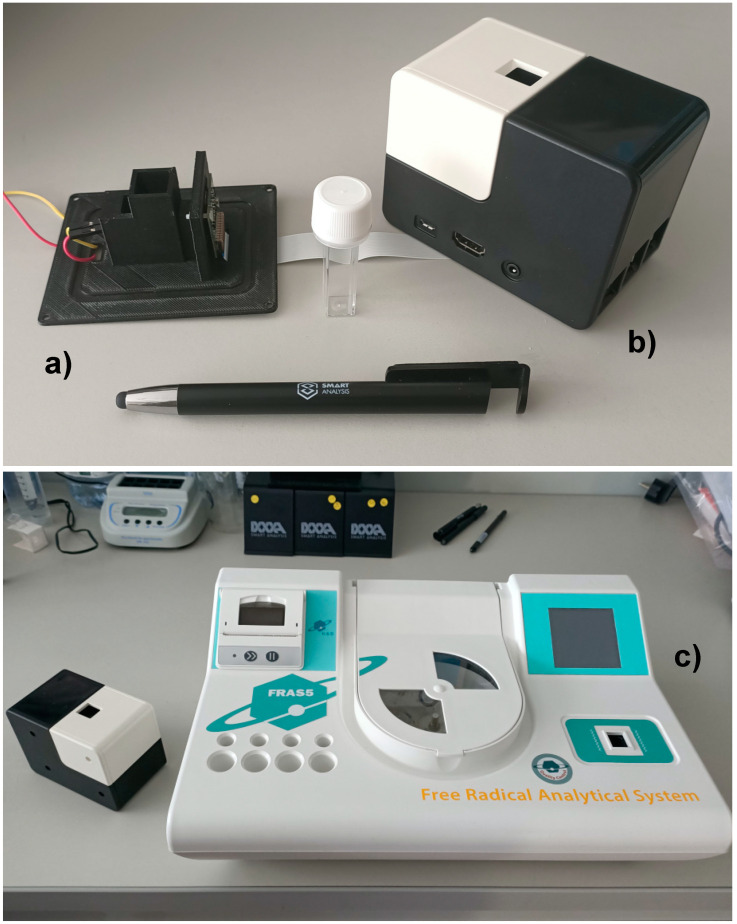
Setups employed for the analysis. (**a**) The 3D-printed device that accommodated the PiCam, the cuvette holder, the light diffuser and the LED. (**b**) We-Lab device (DNAPhone srl). (**c**) The FRAS5 instrument for the oxidative stress measurement (H&D SRL).

**Figure 4 sensors-21-03552-f004:**
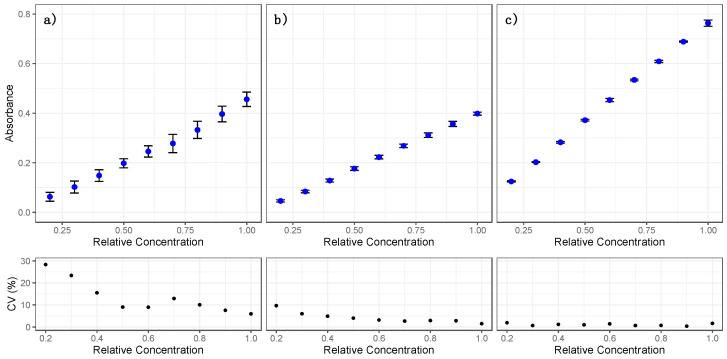
Reading of dilutions of amaranth dye with the 3D-printed Raspberry Pi-based device. (**a**) Automatic mode (AM); (**b**) manual mode (MM); (**c**) raw mode (RM). Each test consists of three replicas. The bottom part of each image reports the relative coefficient of variation (CV%).

**Figure 5 sensors-21-03552-f005:**
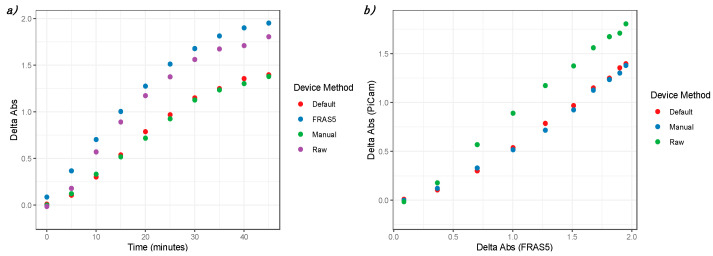
(**a**) Kinetic characterization of the d-ROM kit reaction by the Raspberry Pi-based device We-Lab using AM, MM, RM modes. FRAS5 (H&D srl) was used as reference system. (**b**) Data correlation between We-Lab and FRAS5.

**Figure 6 sensors-21-03552-f006:**
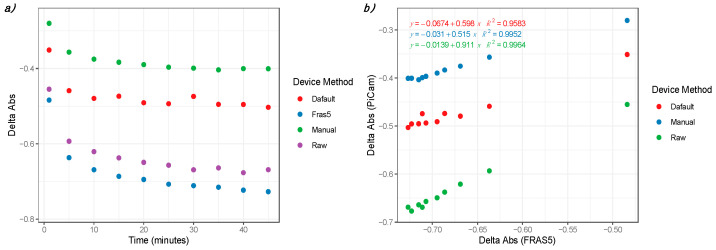
(**a**) Kinetic characterization of the PAT kit reaction by the Raspberry Pi-based device We-Lab using AM, MM, RM modes. FRAS5 (H&D srl) was used as reference system. (**b**) Data correlation between We-Lab and FRAS5.

**Figure 7 sensors-21-03552-f007:**
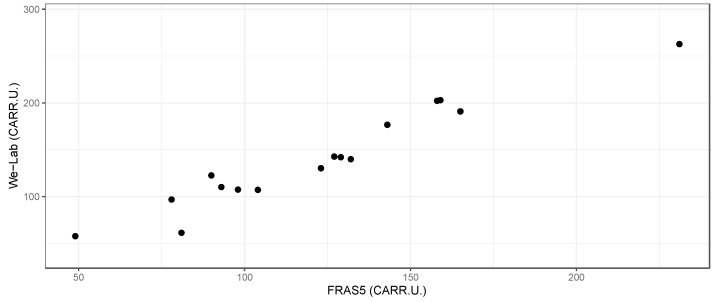
Panel of horse blood samples analyzed using d-ROM kit reaction by the Raspberry Pi-based device We-Lab set in RM modes compared with FRAS5 (H&D SRL).

## Data Availability

Not applicable.

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
