# Peer review of "Sensing Optimum in the Raw: Leveraging the Raw-Data Imaging Capabilities of Raspberry Pi for Diagnostics Applications"

_sensors, 2021, doi:10.3390/s21103552_

Round 1
Reviewer 1 Report
This manuscript concern an interesting hardware/software research operating micro-computers and sensing technologies setup for diagnostic applications. The work proposed has the benefit of being based on open and low-cost system and establish a prominent comparison with alternative system, including commercial solutions. The context is efficiently introduced and the study is well-conducted all over the paper. All explanation and details are given to ease the reading and the understanding including for non-expert readers.
Given a point of view from outside the specificity of this domain and application field using sensing technology, this paper provides a strong impression of a solid, rigorous and serious work. The development is limpid and each section brings necessary analytics plot to support the results. The litterature is rich and relevant.
The fact that the proposed system (open and low-cost) could improve the educational and dissemination purposes is a very valuable added-value to this work.
If pertinent the proposal could be enrich by extending the discussion on :
- the raw image processing and colorimetric calibration (only because I'm coming from a domain where this point a top list priority ). The authors indeed make the hypothesis of alteration of signal because of image processing, it would be valuable for other field to deepen this question.
- pi-cam systems while a lot of applications suffer from low quality of optical system compatible
- additional insights that could serve or come from other domain operating raspi-based imaging (photometric stereo, Reflectance Transformation Imaging etc)
As it seems that authors are devoted to open-hardware/software philosophy (despite of commercial purposes) it will be valuable to state, if, how and where all or part the system (from raw decoder to complete 3D printed setup) will be accessible (for educational and research uses).
Very few additional comments within the annotated PDF.

Reviewer 2 Report
Comments to the Review Type of Manuscript entitled "Sensing optimum in the raw: leveraging the raw-data imaging capabilities of Raspberry Pi for diagnostics applications. " I will start by congratulating the authors for this exciting and valuable work. They show the enormous potential of future miniaturized instrumentation based on assembled commercial technology that could perform at reasonable levels compared to scientific instrumentation. Moreover, the manuscript's content is of extreme interest, in particular, for the sensors field. Although I am pretty satisfied with the presented content of the manuscript, I do have some requests before the final acceptance of the manuscript. Therefore, the authors must provide: a) the algorithms used in all experiments, with a detailed explanation, exclusively in section 2.1. b) the concentrations and all the spectra acquired with the UV-Vis spectrophotometer. c) A more detailed explanation about the ROI selection, and if this selection is already automated, the corresponding algorithm. If the ROI selection is NOT YET automated, they should discuss the potential drawback of this issue in terms of time-cost data analysis compared with the scientific instrumentation. d) In section 2.6, the absorbance equation shows the residual intensity of the excitation light source. Could the authors avoid this residual signal? e) There are a few typos, like the caption of Figure 2, such as "set-ups". Please, check all of them and corrected them. After the shortlist of issues mentioned above being addressed and well addressed, I will recommend the acceptance of the manuscript.
